# Symmetry Detection and Breaking in Cost-Optimal Numeric Planning

**Alexander Shleyfman[1], Ryo Kuroiwa[2], J. Christopher Beck[2]**

[1]The Faculty of Industrial Engineering and Management, Technion, Haifa, Israel
[2]Department of Mechanical and Industrial Engineering, University of Toronto, Toronto, Canada
shleyfman.alexander@gmail.com, ryo.kuroiwa@mail.utoronto.ca, jcb@mie.utoronto.ca

## Abstract

One of the main difficulties affecting the use of domain-independent numeric planning is the complexity of the search problem. The exploitation of structural symmetries in a search problem can constitute an effective method of pruning search branches that may lead to exponential improvements in performance. Over a decade now, symmetry breaking techniques have been successfully used within both optimal and satisficing classical planning. In this work, we show that symmetry detection methods applied in classical planning with some effort can be modified to detect symmetries in linear numeric planning. The detected symmetry group, thereafter, can be used almost directly in the $A^*$-based symmetry breaking algorithms such as DKS and Orbit Space Search. We empirically validate that symmetry pruning can yield a substantial reduction in the search effort, even if algorithms are equipped with a strong heuristic, such as LM-cut.

## Introduction

Deterministic planning is the problem of finding a sequence of actions that brings the actor from a given to some desired state. While the formalisms to describe this paradigm may vary, it seems reasonable to assume that richer models can capture finer aspects, and thus, represent the problem with higher fidelity. For example, in classical planning, the variables of the problem are restricted to finite domains, whereas the numeric variant of planning encompasses both continuous and finite variable ranges. Satisficing planners that can manage numeric fluents were designed at the beginning of the century (Hoffmann 2003), yet, it seems that the progress was slowed due to the theoretical undecidability of even simplest, yet still meaningful, numeric formalisms (Helmert 2002). In recent years, however, there seems to have been a surge of interest in planning with numeric fluents, resulting in the development of multiple heuristics for both optimal and satisficing settings (Aldinger and Nebel 2017; Scala et al. 2016; Scala, Haslum, and Thiébaux 2016; Scala et al. 2017; Piacentini et al. 2018; Scala et al. 2020; Kuroiwa et al. 2021). Unfortunately, having a good heuristic is not enough to assemble an efficient planner, e.g., $A^*$ can expand an exponential number of states even when equipped with an almost perfect heuristic (Helmert and Röger 2008).

Partially to account for this deficit, pruning methods were developed for classical planning (Fox and Long 2002; Coles and Coles 2010; Nissim, Apsel, and Brafman 2012; Wehrle and Helmert 2012; Holte and Burch 2014), and in the past decade, the use of symmetry-based pruning methods has shown its potential within the context of forward search (Pochter, Zohar, and Rosenschein 2011; Domshlak, Katz, and Shleyfman 2013; Wehrle et al. 2015; Gnad et al. 2017). In particular, symmetry reduction methods such as DKS and Orbit Space Search (OSS) were effectively applied in a wide range of classical planning domains, often substantially reducing the expanded state-space size, with a significant increase in planning performance (Domshlak, Katz, and Shleyfman 2012, 2015). In classical planning, symmetry reduction methods compute equivalence classes of states, where the equivalence relation of these classes is based on a precomputed symmetry group. The search exploits these classes by replacing all states in this class with some representative state. Domshlak et al. have shown that given a "path" where each consequent state was replaced by a representative state, one may efficiently reconstruct a corresponding path in the original state space. Hence, the expanded search tree must contain at most one representative of each class at all times.

In this work, we show that the graph-based symmetry detection method proposed by Pochter et al. can be adapted for the numeric setting. We extend the notion of structural symmetries proposed by Shleyfman et al. (2015) to account for linear numeric formulas and demonstrate the equivalence of the obtained symmetries. We also established that from the theoretical standpoint computing these numeric symmetries is not harder than computing the symmetries for classical planning. By grounding these symmetries to the state space level, we enable the use of both DKS and OSS practically as is. Finally, our experimental evaluation demonstrates that in presence of symmetries in the planning task the symmetry breaking algorithms compete favorably with $A^*$, even if equipped with a strong heuristic such as numeric LM-cut.

## Preliminaries

We consider a fragment of numeric planning restricted to the FDR formalism (Bäckström and Klein 1991; Bäckström and Nebel 1995; Helmert 2009) with the addition of numeric state variables, where the conditions and effects on numeric variables are restricted to linear formulas, and unsurprisingly called *linear numeric planning* task (LT). Formally, LT is de-

fined as a 4-tuple $\Pi = \langle \mathcal{V}, \mathcal{A}, s_I, G \rangle$, where $\mathcal{V} = \mathcal{V}_p \cup \mathcal{V}_n$, with $\mathcal{V}_p$ is being a finite sets of propositional variable, where each variable $v \in \mathcal{V}_p$ is associated with a finite domain of values $\mathcal{D}(v)$, and $\mathcal{V}_n$ is a set of numeric variables. Numeric variables $v \in \mathcal{V}_n$ have rational values, i.e., $\mathcal{D}(v) = \mathbb{Q}$. If $\Pi$ does not have any numeric state variable ($\mathcal{V}_n = \emptyset$), we have a *classical* (FDR) *planning task* $\Pi_{FDR}$. Assignment of a valid value $d$ to a variable $v \in \mathcal{V}$ is called a fact and denoted by $\langle v, d \rangle$. For a subset of variables $V \subseteq \mathcal{V}$ we define its joint domain to be $\mathcal{D}[V] = \times_{v \in V} \mathcal{D}(v)$. The state space is $\mathcal{S} = \mathcal{D}[\mathcal{V}]$. A state $s \in \mathcal{S}$ is a full assignment over all variables, and can be seen as a tuple $\langle s_p, s_n \rangle$, where $s_p \in \mathcal{D}[\mathcal{V}_p]$ and $s_n \in \mathcal{D}[\mathcal{V}_n]$; $s[v]$ indicates the value of the variable $v \in \mathcal{V}$ over the state $s$. $s_I$ is a state. Note that $s$, $s_p$, and $s_n$ may be presented via vector representation, or as a set of facts $s = s_p \cup s_n$. In the latter case, there are no two facts that involve the same variable, and $|s| = |\mathcal{V}|$. The value of a variable $v$ in $s$ is given by $s[v] = d$, and is equivalent to $\langle v, d \rangle \in s$. A partial state is a subset $s^{pt} \subseteq s$ of some $s \in \mathcal{S}$.

A *linear expression* $\xi$ has the form $\xi = \sum_{v \in V} w_v^\xi v + w_0^\xi$, where $V \subseteq \mathcal{V}_n$, with $\forall v \in V, w_v^\xi \in \mathbb{Q}$, and $w_0^\xi \in \mathbb{Q}$. The value of $\xi$ in a state $s$ is given by the expression $s[\xi] = \sum_{v \in V} w_v^\xi s[v] + w_0^\xi$. For simplicity, we assume that there is always a variable $v_0 \in \mathcal{V}_n$ such that for each state $s$ it holds $s[v_0] = 1$. This assumption allows a more convenient expression $\xi = \sum_{v \in V} w_v^\xi v$. We assume that there are no redundant factors in the condition representation, i.e., for all $v \in V$ it holds $w_v \neq 0$. $\Xi$ is the set of all linear expressions in $\Pi$. The set of all constants that appear in $\xi$ is denoted by $nums(\xi)$, and the set of all variables that appear in $\xi$ is denoted by $vars(\xi)$. Similarly, for a partial state $s^{pt}$ we denote by $vars(s^{pt}) \subseteq \mathcal{V}$ the variables involved in $s^{pt}$.

Conditions can be either propositional or numeric. A propositional condition $\psi$ is a partial state over the variables $vars(\psi) \subseteq \mathcal{V}_p$. We say that $\psi$ is satisfied by $s$, $s \models \psi$, if $\psi \subseteq s_p$. A propositional condition $\psi$ has the form $\xi \geq 0$, where $\xi$ is a linear expression. In this case, we say that $\psi$ is satisfied by $s$ if $\xi[s] \geq 0$ holds. The set of conditions $\Psi$ is satisfied by $s$, if for each $\psi \in \Psi$ it holds that $s \models \psi$. The goal condition $G = G_p \cup G_n$ is a union over sets of propositional and numeric conditions, respectively.

Action $a = \langle \mathsf{pre}(a), \mathsf{eff}(a), \mathsf{cost}(a) \rangle \in \mathcal{A}$ has preconditions $\mathsf{pre}(a) = \mathsf{pre}_p(a) \cup \mathsf{pre}_n(a)$, effects $\mathsf{eff}(a) = \mathsf{eff}(a)_p \cup \mathsf{eff}_n(a)$, and cost $\mathsf{cost}(a) \in \mathbb{R}^{0+}$. $\mathsf{pre}_p(a)$ is a partial state over propositional variables and $\mathsf{pre}_n(a)$ is a set of linear numeric conditions. The set of all numeric conditions of the task is denoted by $\Psi_n$, i.e., $\Psi_n = G_n \cup \bigcup_{a \in \mathcal{A}} \mathsf{pre}_n(a)$. $a$ is applicable to $s$ if $s \models \mathsf{pre}(a)$. Similarly, the propositional effect $\mathsf{eff}_p(a)$ is a partial state on a subset of $\mathcal{V}_p$. The effect $\mathsf{eff}_n(a)$ is a set of numeric effects of the form $(v \mathrel{+}= \xi)$, where $v \in \mathcal{V}_n$ and the value of $\xi \in \Xi$. We assume that assignment effect $v := \xi$ and subtractive effect $v \mathrel{-}= \xi + c$ are normalized to the additive forms $v \mathrel{+}= \xi - v$ and $v \mathrel{+}= -\xi - c$, and one action has at most one effect on the same numeric variable. The result of applying $a$ in $s$ is denoted by $s[\![a]\!] = s_p' \cup s_n'$, where the resulting state is defined as $s_p'[v] = \mathsf{eff}_p(a)[v]$ for $v \in vars(\mathsf{eff}_p(a))$, $s[\![a]\!][v] =$

$s[v] + \xi[s]$ if $(v \mathrel{+}= \xi) \in \mathsf{eff}_n(a)$, and $s[\![a]\!][v] = s[v]$ otherwise. $\mathcal{A}(s)$ is the set of all actions applicable to $s$.

An *s-plan* is an action sequence $\pi$ that can be applied successively in $s$ and results in a goal state $s_* \models G$. A plan for $\Pi$ is an $s_I$-plan. The *cost of an s-plan* $\pi$ is the sum of all its action costs and an *optimal s-plan* has minimal cost among all possible $s$-plans.

A *state transition graph* is a labeled digraph $\mathcal{T}_\Pi = \langle \mathcal{S}, E \rangle$, whose vertexes $\mathcal{S}$ are the states of $\Pi$, the set of labeled arcs $E = \{\langle s, s[\![a]\!]; a \rangle \mid s \in \mathcal{S}, a \in \mathcal{A}(s)\}$ is induced by the actions of $\Pi$, where $\mathsf{cost}(s, s[\![a]\!]; a) = \mathsf{cost}(a)$. A plan for $\Pi$ is equivalent to a path from $s_I$ to $s_*$ in $\mathcal{T}_\Pi$.

## Structural Symmetries and PDG

This subsection defines the notion of *structural symmetries* (Shleyfman et al. 2015), which captures previously proposed concepts of symmetries in classical planning. In short, structural symmetries relabel a given planning task. Variables are mapped to variables, values to values (preserving the $\langle var, val \rangle$ structure), and actions are mapped to actions. In this work, we follow the definition of structural symmetries for FDR planning tasks as defined by Wehrle *et al.* (2015). For a planning task $\Pi_{FDR} = \langle \mathcal{V}, \mathcal{A}, I, G \rangle$, let $P$ be the set of $\Pi$'s facts, and let $P_\mathcal{V} := \{\{\langle v, d \rangle \mid d \in \mathcal{D}(v)\} \mid v \in \mathcal{V}\}$ be the set of sets of facts attributed to each variable in $\mathcal{V}$. We say that a permutation $\sigma : P \cup \mathcal{A} \to P \cup \mathcal{A}$ is a *structural symmetry* if the following holds:

1. $\sigma(P_\mathcal{V}) = P_\mathcal{V}$,
2. $\sigma(\mathcal{A}) = \mathcal{A}$, and, for all $a \in \mathcal{A}$, $\sigma(\mathsf{pre}(a)) = \mathsf{pre}(\sigma(a))$, $\sigma(\mathsf{eff}(a)) = \mathsf{eff}(\sigma(a))$, and $\mathsf{cost}(\sigma(a)) = \mathsf{cost}(a)$.
3. $\sigma(G) = G$.

We define the application of $\sigma$ to a set $X$ by $\sigma(X) := \{\sigma(x) \mid x \in X\}$, where $\sigma$ is applied recursively up to the level of action labels and facts. For example, let $s$ be a partial state, since $s$ can be represented a set of facts, applying $\sigma$ to $s$ results in a partial state $s'$, s.t. for all facts $\langle v, d \rangle \in s$ it holds that $\sigma(\langle v, d \rangle) = \langle \sigma(v), d' \rangle \in s'$ and $s'[\sigma(v)] = d'$. This implies that $\sigma$ uniquely defines the values of each variable, i.e., if $d \in \mathcal{D}(v)$, we can write $\sigma(d) = d'$, where $d' \in \mathcal{D}(\sigma(v))$.

Note that the set of all structural symmetries of an FDR task is a finite set of bijections closed under composition. Thus, we have that structural symmetries form a group over the task $\Pi_{FDR}$, denoted by $Aut(\Pi_{FDR})$.

## Symmetries and Problem Description Graphs

The problem description graph (PDG) was introduced by Pochter *et al.* (2011), and later on reformulated by Domshlak *et al.* (2012), and Shleyfman *et al.* (2015).

**Definition 1.** *Let* $\Pi_{FDR}$ *be a* FDR *planning task. The **problem description graph** $PDG_{\Pi_{FDR}}$ is the colored digraph* $\langle N, E, \mathsf{col} \rangle$ *with nodes*

$$N = N_\mathcal{V} \cup \bigcup_{v \in \mathcal{V}} N_{\mathcal{D}(v)} \cup N_\mathcal{A}$$

*where* $N_\mathcal{V} = \{n_v \mid v \in \mathcal{V}\}$, $N_{\mathcal{D}(v)} = \{n_v^d \mid d \in \mathcal{D}(v)\}$, *and*

$N_{\mathcal{A}} = \{n_a \mid a \in \mathcal{A}\}$; *node colors*

$$\mathsf{col}(n) = \begin{cases} 0 & \text{if } n \in N_{\mathcal{V}} \\ 1 & \text{if } n_v^d \in \bigcup_{v \in \mathcal{V}} N_{\mathcal{D}(v)} \wedge \langle v, d \rangle \in G \\ 2 & \text{if } n_v^d \in \bigcup_{v \in \mathcal{V}} N_{\mathcal{D}(v)} \wedge \langle v, d \rangle \notin G \\ 3 + \mathsf{cost}(a) & \text{if } n_a \in N_{\mathcal{A}} \end{cases}$$

*and edges*

$$E = \bigcup_{v \in \mathcal{V}} E^v \cup \bigcup_{a \in \mathcal{A}} E_a^{\mathsf{pre}} \cup E_a^{\mathsf{eff}},$$

*where* $E^v = \{(n_v, n_v^d) \mid d \in \mathcal{D}(v)\}$, $E_a^{\mathsf{pre}} = \{(n_a, n_v^d) \mid \langle v, d \rangle \in \mathsf{pre}(a)\}$, *and* $E_a^{\mathsf{eff}} = \{(n_v^d, n_a) \mid \langle v, d \rangle \in \mathsf{eff}(a)\}$.

In their work, Pochter *et al.* observed that *PDG symmetry* is a symmetry of $\mathcal{T}_{\Pi}$ that is induced by a graph automorphism of the *PDG* of $\Pi_{FDR}$.[1] In what follows, we will denote by $Aut(PDG(\Pi_{FDR}))$ the automorphism group of the PDG of the task $\Pi_{FDR}$. Shleyfman *et al.* (2015), in turn showed that every structural symmetry of $\Pi_{FDR}$ corresponds to a PDG symmetry in the sense that they induce the same transition graph symmetry, i.e., the groups $Aut(PDG(\Pi_{FDR}))$ and $Aut(\Pi_{FDR})$ are isomorphic, namely $Aut(PDG(\Pi_{FDR})) = Aut(\Pi_{FDR})$.

Group *homomorphism f* is a function between two groups $f : \Gamma \to \hat{\Gamma}$ that respects the group operations. i.e., given $\sigma, \sigma' \in \Gamma$ it holds $f(\sigma \circ \sigma') = f(\sigma) \circ f(\sigma')$. Isomorphism is a bijective (one-to-one and onto) homomorphism. Note that all isomorphsims are invertible, where the inverse is also an isomorphism. When the groups $\Gamma$ and $\hat{\Gamma}$ are isomorphic, we write $\Gamma = \hat{\Gamma}$. If homomorphism $f : \Gamma \to \hat{\Gamma}$ is injective (one-to-one) we say that $\Gamma$ is a subgroup of $\hat{\Gamma}$ and write $\Gamma \le \hat{\Gamma}$.

### Symmetries of the State Transition Graph

A *symmetry* of a transition graph $\mathcal{T}_{\Pi} = \langle \mathcal{S}, E \rangle$ with actions $\mathcal{A}$ is a permutation $\sigma$ of $\mathcal{S} \cup \mathcal{A}$ mapping states to states and actions to actions such that
- $\langle s, s'; a \rangle \in E$ iff $\langle \sigma(s), \sigma(s'); \sigma(a) \rangle \in E$,
- $\mathsf{cost}(\sigma(a)) = \mathsf{cost}(a)$, and
- $s$ is a goal state iff $\sigma(s)$ is a goal state

for all states $s$, $s'$ and actions $a$. Symmetries are also called *(goal-stable) automorphisms*. They are closed under composition and inverse, forming the *automorphism group* $Aut(\mathcal{T}_{\Pi})$ of the transition graph. Each subgroup $\Gamma$ of symmetries induces an equivalence relation $\sim_{\Gamma}$ on states $\mathcal{S}$: $s \sim_{\Gamma} s'$ iff $\sigma(s) = s'$ for some $\sigma \in \Gamma$. States in the same equivalence class are called *symmetric*.

The following (immediate) result is the formal basis for exploiting symmetries for planning:

**Theorem 1.** *Let $\Pi$ be a planning task, let $s$ be one of its states, let $\pi$ be a sequence of actions of $\Pi$, and let $\sigma$ be a symmetry of $\mathcal{T}_{\Pi}$. Then $\pi$ is a plan for $s$ iff $\sigma(\pi)$ is a plan for $\sigma(s)$, and the two plans have the same cost.*

Note that the definition of symmetries of a state transition graph depends only on the notion of actions applicable to

---

[1]The formal proof can be found in Shleyfman (2020).

states, $s' = s[\![a]\!]$, and the notion of goal state, thus it fits multiple formalisms that support this dynamic. Particularly, this definition match not only the transition graph induced by an FDR task but also the one induced by an LT.

A $PDG_{\Pi_{FDR}}$ *symmetry* is a symmetry of $\mathcal{T}_{\Pi_{FDR}}$ that is induced by a color preserving graph automorphism of $PDG_{\Pi_{FDR}}$. The group $Aut(PDG(\Pi_{FDR})) = Aut(\Pi_{FDR})$ induces a subgroup of $Aut(\mathcal{T}_{\Pi_{FDR}})$, which in turn defines an equivalence relation over the states $S$ of $\Pi_{FDR}$. In the following section we define a numeric versions of PDG and structural symmetries that obey the same relation, i.e., induce a symmetry group of the state transition graph.

## Symmetries in Numeric Domains

Shleyfman and Jonnson (2021) show that determining whether two states are symmetric in a transition system induced by an FDR planning task is PSPACE-hard. This result automatically grants us that PSPACE-hardness also holds for numeric planning, since it, trivially, contains the FDR formalism. What is more troublesome, is the fact that the presence of numeric fluents makes the search space infinite, which may lead to a lot of unpleasant properties of its symmetry group. Consider, for example, that the permutation group of a countable number of elements, contains an uncountable number of symmetries, infinitely many of which have infinite order.

Since identifying the whole symmetry group of the transition task is infeasible, we would like to obtain a manageable subgroup and use it for symmetry breaking in the search over the state space. To this end, we extend the Structural Symmetries and the *PDG* by numeric fluents. Then, we show that Numeric Structural Symmetries (NSS) can be mapped into symmetries of the state transition graph (Thm. 2), and that the symmetry group of the numeric version of *PDG* is isomorphic to the NSS group of the task (Thm. 3). For the flowchart of the proof of the grounding process see Figure 1.

We exploit the grounded, effectively computed subgroup $\Gamma_{\Pi}$ of $Aut(\mathcal{T}_{\Pi})$ by plugging it into the symmetry breaking searches DKS (Domshlak, Katz, and Shleyfman 2012), OSS (Domshlak, Katz, and Shleyfman 2015).

Let us give here a motivational example. Suppose we have a planning task where homogeneous trucks deliver various cargo across a city, where the city map is represented via digraph. A truck $T$ in this task is represented with three variables: $loc(T)$ the locations of the truck (finite domain variable), $fuel(T)$ the amount of fuel in the truck, and $load(T)$ the current load of the truck (both numeric variables). Since the trucks are homogeneous, we would like to have a symmetry $\sigma$ to capture this information, i.e., given two trucks $T_1$ and $T_2$ a map that replaces only the labels of the trucks should induce an automorphism of the transition graph. In what follow we would like to capture this behavior. Instead of saying that $T_1$ is symmetric to $T_2$, we want a bijection that switches between the variables associated with $T_1$ and $T_2$, as follows: $\sigma(loc(T_1)) = loc(T_2)$, $\sigma(fuel(T_1)) = fuel(T_2)$, $\sigma(load(T_1)) = load(T_2)$, and vise versa.

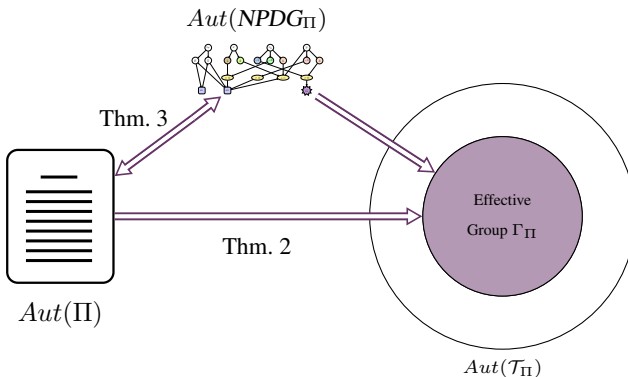

Figure 1: Schematic representation of detection and grounding of structural symmetries for linear numeric planning.

## Numeric Structural Symmetries

For an LT $\Pi = \langle \mathcal{V}, \mathcal{A}, I, G \rangle$. We say that a permutation $\sigma$ over $\mathcal{V} \cup \mathcal{A} \cup \left( \bigsqcup_{v \in \mathcal{V}_p} \mathcal{D}(v) \right)$ is a *numeric structural symmetry (NSS)* if the following holds:[2]

1. $\sigma(\mathcal{V}_p) = \mathcal{V}_p$, $\sigma(\mathcal{V}_n) = \mathcal{V}_n$,
2. for all $v \in \mathcal{V}_p$ holds $\sigma(\mathcal{D}(v)) = \mathcal{D}(\sigma(v))$,
3. $\sigma(\mathcal{A}) = \mathcal{A}$.

We define an application of $\sigma$ to a partial state $s^{pt}$ over the propositional variables $vars(s^{pt})$, as $\sigma(s^{pt})[\sigma(v)] := \sigma(s^{pt}[v])$ for all $v \in vars(s^{pt})$. Note that by points 1 and 2 the result of this application, $\sigma(s^{pt})$, is also a partial state.

Let $\xi = \sum_{v \in V} w_v^{\xi} v$ be a linear formula over the variables $V \subseteq \mathcal{V}_n$. We define the application of $\sigma$ to $\xi$ to be $\sigma(\xi) = \sum_{v \in V} w_v^{\xi} \sigma(v)$. The application of $\sigma$ to condition $\xi \geq 0$ is defined as $\sigma(\xi) \geq 0$. The application of $\sigma$ to numeric effect $v \mathrel{+}= \xi \in \mathsf{eff}_n(a)$ is written as $\sigma(v) \mathrel{+}= \sigma(\xi)$, where $v \in \mathcal{V}_n$. Using these notations we establish the following conditions on $\sigma$.

4. for all $a \in \mathcal{A}$, $\sigma(\mathsf{pre}_n(a)) = \mathsf{pre}_n(\sigma(a))$, $\sigma(\mathsf{pre}_p(a)) = \mathsf{pre}_p(\sigma(a))$, $\sigma(\mathsf{eff}_p(a)) = \mathsf{eff}_p(\sigma(a))$, $\sigma(\mathsf{eff}_n(a)) = \mathsf{eff}_n(\sigma(a))$, and $\mathsf{cost}(\sigma(a)) = \mathsf{cost}(a)$.
5. $\sigma(G_n) = G_n$ and $\sigma(G_p) = G_p$.

Note that immediate consequence of this definition is that for any structural symmetry $\sigma$, it holds that $\sigma(\Xi) = \Xi$.

Note also that that for two structural symmetries $\sigma_1, \sigma_2$ the composition $\sigma_1 \circ \sigma_2$ is also a structural symmetry.

- This is straightforward for properties 1, 3, and 5: $\sigma_1 \circ \sigma_2(X) = \sigma_1(\sigma_2(X)) = \sigma_1(X) = X$ where $X \in \{\mathcal{V}_n, \mathcal{V}_p, \mathcal{A}, G_n, G_p\}$.
- Property 4 holds for each $a \in \mathcal{A}$ since for each $h \in \{\mathsf{pre}_n, \mathsf{pre}_p, \mathsf{eff}_n, \mathsf{eff}_p, \}$ we have $\sigma_1 \circ \sigma_2(h(a)) = \sigma_1(\sigma_2(h(a))) = \sigma_1(h(\sigma_2(a))) = h(\sigma_1(\sigma_2(a))) = h(\sigma_1 \circ \sigma_2(a))$. By replacing $a$ by $v \in \mathcal{V}_p$ and $h$ by $\mathcal{D}$, we get property 2. The check for cost is trivial and repeats the previous point.

---

[2]Example of disjoint union: $\{5\} \sqcup \{5\} = \{\langle 5, 0 \rangle, \langle 5, 1 \rangle\}$.

A well-known result in group theory (Herstein 1975) states that a finite set closed under an operation forms a group. Here we denote the group of all structural symmetries as $Aut(\Pi)$. This definition leads us to the observation below.

**Theorem 2.** *There is a natural injection from $Aut(\Pi)$ to $Aut(\mathcal{T}_\Pi)$, i.e., for each $\sigma \in Aut(\Pi)$ there is a unique $\tilde{\sigma} \in Aut(\mathcal{T}_\Pi)$.*

*Proof.* Let $\mathcal{T}_\Pi = \langle \mathcal{S}, E \rangle$ be a transition system induced by a LT $\Pi = \langle \mathcal{V}, \mathcal{A}, s_I, G \rangle$, and let $\sigma \in Aut(\Pi)$ be a structural symmetry. For a state $s = s_p \cup s_n \in \mathcal{S}$ we define the application of $\tilde{\sigma}$ to $s$ as $\tilde{\sigma}(s_p \cup s_n) = \tilde{\sigma}(s_p) \cup \tilde{\sigma}(s_n)$, where

$$\tilde{\sigma}(s_p) = \{\sigma(\langle v, d_v \rangle) \mid \langle v, d_v \rangle \in s_p\} = \{\langle \sigma(v), \sigma(d_v) \rangle \mid \langle v, d_v \rangle \in s_p\},$$

$\tilde{\sigma}(s_n) = \{\langle \sigma(v), q_v \rangle \mid \langle v, q_v \rangle \in s_n\}$ and $\tilde{\sigma}(a) = \sigma(a)$ for $a \in \mathcal{A}$. By properties 1-3, we have that $\tilde{\sigma}$ is a permutation over $\mathcal{S} \cup \mathcal{A}$, and the map $\sigma \mapsto \tilde{\sigma}$ is injective.

Let $\langle s, s'; a \rangle \in E$, and let $\sigma$ be a structural symmetry. We aim to show that $\langle \tilde{\sigma}(s), \tilde{\sigma}(s'); \tilde{\sigma}(a) \rangle \in E$. Let us decouple the states $s$ and $s'$ to their propositional and numeric components $s_p \cup s_n$ and $s'_p \cup s'_n$, respectively. The proof of the propositional part was already done by Shleyfman et al. (2020), but we present it here for the sake of completeness. Note that by definition of $\tilde{\sigma}$ we have $\tilde{\sigma}(s)_p = \sigma(s_p)$. Let $\psi$ be a partial assignment over the propositional variables such that $s_p \models \psi$, note that set wise it can be written as $\psi \subseteq s_p$. By applying $\sigma$ to both sides we have $\sigma(s_p) \models \sigma(\psi)$, thus $\tilde{\sigma}(s) \models \sigma(\psi)$. This claim grants us: $s \models \mathsf{pre}_p(a)$ implies that $\tilde{\sigma}(s) \models \mathsf{pre}_p(\tilde{\sigma}(a))$, and $s' \models \mathsf{eff}_p(a)$ implies that $\tilde{\sigma}(s') \models \mathsf{eff}_p(\tilde{\sigma}(a))$. Since $\sigma$ is a permutation over $\mathcal{V}_p$, its application to the unaffected variables $\mathcal{V}_p \setminus vars(\mathsf{eff}_p(a))$ can be written as $\mathcal{V}_p \setminus vars(\mathsf{eff}_p(\tilde{\sigma}(a)))$, and since $s'[v] = s[v]$ for each $v \in \mathcal{V}_p \setminus vars(\mathsf{eff}_p(a))$ we have that $\tilde{\sigma}(s')[v] = \tilde{\sigma}(s)[v]$ for each $v \in \mathcal{V}_p \setminus vars(\mathsf{eff}_p(\tilde{\sigma}(a)))$, resulting in $\tilde{\sigma}(s)[\![\tilde{\sigma}(a)]\!]_p = \tilde{\sigma}(s')_p$. Note, that the fact that for any partial assignment $\psi$, it holds $s_p \models \psi \iff \tilde{\sigma}(s)_p \models \sigma(\psi)$ which directly implies that $s_p \models G_p \iff \tilde{\sigma}(s)_p \models G_p$.

We aim to show that $\tilde{\sigma}(s)[\![\tilde{\sigma}(a)]\!]_n = \tilde{\sigma}(s')_n$. Let $\psi \in \Psi_n$ be a numeric condition $\psi : \sum_{v \in V} v \cdot w_v^{\psi} \geq 0$ such that $s_n \models \psi$. By definition of $\tilde{\sigma}$ we have $s[v] = \tilde{\sigma}(s)[\sigma(v)]$, thus

$$\sum_{v \in V} s[v] \cdot w_v^{\psi} = \sum_{v \in V} \tilde{\sigma}(s)[\sigma(v)] \cdot w_v^{\psi} \geq 0 \text{ implies that}$$

$$s_n \models \psi \iff \tilde{\sigma}(s)_n \models \sigma(\psi).$$

This statement combined with the previous paragraph has two immediate consequences:
1. $s \models G \iff \tilde{\sigma}(s) \models G$, and
2. $s \models \mathsf{pre}(a) \iff \tilde{\sigma}(s) \models \mathsf{pre}(\sigma(a))$.

Now, to show that $\langle \tilde{\sigma}(s), \tilde{\sigma}(s'); \tilde{\sigma}(a) \rangle \in E$, it is enough ensure that for any $v \in \mathcal{V}_n$ it holds that $s'[v] = \tilde{\sigma}(s')[\sigma(v)]$. Here we have two cases, either $v$ is affected by $\mathsf{eff}_n(a)$ or not. If $v$ is not affected, i.e., $v \notin vars(\mathsf{eff}_n(a))$, then since $\sigma$ is a permutation it holds that $\sigma(v)$ is not affected by $\mathsf{eff}_n(\sigma(a))$, i.e., $\sigma(v) \notin vars(\mathsf{eff}_n(\sigma(a)))$. Thus, $s'[v] = s[v] = \tilde{\sigma}(s)[\sigma(v)] = \tilde{\sigma}(s')[\sigma(v)]$. Otherwise, let $v \mathrel{+}= \xi$

be the numeric effect of $a$ on $v$. Let $\xi = \sum_{v' \in V} w_v^\xi v$. By definition we have that $\sigma(v) \mathrel{+}= \sigma(\xi) \in \mathsf{eff}_n(\sigma(a))$. Thus,

$$s'[v] = s[v] + \sum_{v' \in V} w_v^\xi s[v'] = \tilde{\sigma}(s)[\sigma(v)] + $$
$$\sum_{v' \in V} w_v^\xi \tilde{\sigma}(s)[\sigma(v')] = \tilde{\sigma}(s')[\sigma(v)],$$

since $s[v] = \tilde{\sigma}(s)[\sigma(v)]$ for each $v \in \mathcal{V}_n$ by definition of $\tilde{\sigma}$.

Hence, we have that $\tilde{\sigma}$ is a symmetry of the transition graph $\mathcal{T}_\Pi$, since it satisfies the following three requirements:
– $\langle s, s'; a \rangle \in E$ iff $\langle \sigma(s), \sigma(s'); \sigma(a) \rangle \in E$,
– $\mathsf{cost}(\sigma(a)) = \mathsf{cost}(a)$, and
– $s$ is a goal state iff $\sigma(s)$ is a goal state. $\qquad\square$

This result establishes that structural symmetries induce transition graph symmetries. Next, we show how to compute these symmetries using a variation of the problem description graph modified for numerical planning.

## Numeric Problem Description Graph

As the state transition graph $\mathcal{T}_\Pi$ of a planning task $\Pi$ is usually too large to be given explicitly, symmetries must be inferred from a compact description. Pochter et al. introduced a method for deducing some symmetries of the planning task from automorphisms of a certain graphical structure, the *problem description graph* (PDG) of $\Pi$. Later, Domshlak et al. slightly modified the definition, mainly to add support for general-cost actions. As observed by Pochter et al., every automorphism of the PDG of $\Pi$ induces an automorphism of $\mathcal{T}_\Pi$, and the former can be found using off-the-shelf tools for the discovery of automorphisms in explicit graphs, such as *bliss* (Junttila and Kaski 2007).

Note that in contrast to the propositional FDR the transition system defined by its numeric counterpart may be infinite. We group numeric elements of conditions and effects into a set of linear formulas. Recall, that $\Xi$ denotes the set of all linear formulas that appear in $\Pi$ (both in conditions and in effects). For each $v \in \mathcal{V}_n$, we define $\mathcal{W}(v) = \{w_v^\xi \in \mathbb{Q} \mid \exists \xi \in \Xi : w_v^\xi \in nums(\xi)\}$, the set of all numeric coefficients associated with $v$. Let $\mathcal{C}_\Pi$ be the set of unique costs of actions and constants in numeric variables in the task $\Pi$, i.e.

$$\mathcal{C}_\Pi = \{\mathsf{cost}(a) \mid a \in \mathcal{A}\} \sqcup \bigcup_{v \in \mathcal{V}_n} \mathcal{W}(v).$$

We define a function $\mathsf{ord} : \mathcal{C}_\Pi \to [|\mathcal{C}_\Pi|]$ to be an order preserving function with respect to the lexicographic order over the set $\mathcal{C}_\Pi$, i.e., for $c_1, c_2 \in \mathcal{C}_\Pi$ holds that $c_1 \le c_2$ implies that $\mathsf{ord}(c_1) \le \mathsf{ord}(c_2)$. The main properties of the function $\mathsf{ord}$ that we are interested in are

1. for $c_1, c_2 \in \mathcal{C}_\Pi$ it holds $c_1 = c_2$ iff $\mathsf{ord}(c_1) = \mathsf{ord}(c_2)$,
2. for each $c \in \mathcal{C}_\Pi$ it holds that $\mathsf{ord}(c) \le |\mathcal{C}_\Pi|$.

In the following definition, the function $\mathsf{ord}$ allows us to distinguish between the task constants while operating within a reasonable number of vertex colors.

**Definition 2.** *Let $\Pi = \langle \mathcal{V}, \mathcal{A}, s_I, G \rangle$ be a LT. The **numeric problem description graph** (NPDG$_\Pi$) of $\Pi$ is the colored digraph $\langle N, E, \mathsf{col} \rangle$ with nodes (e.g., Figure 2)*

$$N = N_{\mathcal{V}_p} \cup N_{\mathcal{V}_n} \cup N_{\mathcal{A}} \cup N_\Xi \cup \{n_G\} \cup \bigcup_{v \in \mathcal{V}} N_{\mathcal{D}(v)}, \ where$$

$N_{\mathcal{V}_x} = \{n_v \mid v \in \mathcal{V}_x\} \ for \ x \in \{p, n\}$,
$N_{\mathcal{A}} = \{n_a, \mid a \in \mathcal{A}\}$,
$\forall v \in \mathcal{V}_p : N_{\mathcal{D}(v)} = \{n_d^v \mid d \in \mathcal{D}(v)\}$,
$\forall v \in \mathcal{V}_n : N_{\mathcal{D}(v)} = \{n_w^v \mid w \in \mathcal{W}(v)\}$,
$N_{\Psi_n} = \{n_\xi^\ge \mid \psi : \xi \ge 0 \in \Psi_n\} \ and$
$N_+ = \{n_\xi^+ \mid \exists \xi \in \Xi, a \in \mathcal{A} : v \mathrel{+}= \xi \in \mathsf{eff}_n(a)\}$,
*with* $N_\Xi = N_{\Psi_n} \cup N_+$, *edges*

$$E = E_{\mathcal{V}_p} \cup E_{\mathcal{V}_n} \cup E_\Xi \cup E_{\mathcal{A}} \cup E_G, \ where$$

$E_{\mathcal{V}_p} = \bigcup_{v \in \mathcal{V}_p} \{(n_v, n_d^v) \mid d \in \mathcal{D}(v)\}$,
$E_{\mathcal{V}_n} = \bigcup_{v \in \mathcal{V}_n} \{(n_v, n_w^v) \mid w \in \mathcal{W}(v)\}$,
$E_\Xi = \bigcup_{n_\xi \in N_\Xi} \{(n_{w_v^\xi}^v, n_\xi) \mid v \in \mathcal{V}_n, w_v^\xi \in nums(\xi)\}$,
$E_{\mathcal{A}} = \bigcup_{a \in \mathcal{A}} \left( E_a^{\mathsf{pre}_n} \cup E_a^{\mathsf{pre}_p} \cup E_a^{\mathsf{eff}_p} \cup E_a^{\mathsf{eff}_n} \right)$, *with*

$E_a^{\mathsf{pre}_n} = \{(n_\xi^\ge, n_a) \mid \psi : \xi \ge 0 \in \mathsf{pre}_n(a)\}$,
$E_a^{\mathsf{pre}_p} = \{(n_d^v, n_a) \mid \langle v, d \rangle \in \mathsf{pre}_p(a)\}$,
$E_a^{\mathsf{eff}_n} = \{(n_a, n_\xi^+), (n_\xi^+, n_v) \mid v \mathrel{+}= \xi \in \mathsf{eff}_n(a)\}$,
$E_a^{\mathsf{eff}_p} = \{(n_a, n_d^v) \mid \langle v, d \rangle \in \mathsf{eff}_p(a)\}$,

$E_G = \{(n_\xi^\ge, n_G) \mid \xi \ge 0 \in G_n\} \cup \{(n_v^d, n_G) \mid \langle v, d \rangle \in G_p\}$, *and node colors*

$$\mathsf{col}(n) = \begin{cases} \mathsf{ord}(\mathsf{cost}(a)) & \textit{if } n = n_a \in N_{\mathcal{A}}, \\ \mathsf{ord}(w) & \textit{if } n = n_w^x \in \bigcup_{v \in \mathcal{V}_n} N_{\mathcal{D}(v)}, \\ |\mathcal{C}_\Pi| + 1 & \textit{if } n \in N_\Xi, \\ |\mathcal{C}_\Pi| + 2 & \textit{if } n = n_G, \\ 0 & \textit{otherwise}. \end{cases}$$

**Theorem 3.** *$Aut(\textit{NPDG}_\Pi)$ and $Aut(\Pi)$ are isomorphic.*

*Proof.* Let $\Pi$ be a numeric planning task, with the corresponding $\textit{NPDG}_\Pi = \langle N, E, \mathsf{col} \rangle$. We start with the definition of the map $f : Aut(\textit{NPDG}_\Pi) \to Aut(\Pi)$ that we aim to prove to be a bijection.

Let $f(\alpha) = l^{-1} \circ \alpha \circ l \in Aut(\Pi)$, where the map

$$l : \mathcal{V} \cup \mathcal{A} \cup \left( \bigsqcup_{v \in \mathcal{V}_p} \mathcal{D}(v) \right) \to N$$

is given by $l(x) \mapsto n_x$, with a slight abuse of notation $l(\langle v, d \rangle) = n_d^v$. Note that $f$ is a homomorphism, i.e., $f(\alpha) \circ f(\beta) = f(\alpha \circ \beta)$.

First, we need to show that $f$ **is well-defined**, i.e., $f(\alpha) \in Aut(\Pi)$. Since $\alpha \in Aut(\textit{NPDG}_\Pi)$ is color-, edge- and degree-preserving we have that the vertex sets $N_{\mathcal{V}_p}, N_{\mathcal{V}_n}, N_{\mathcal{A}}, N_{\Psi_n}, N_+, \bigcup_{v \in \mathcal{V}_p} N_{\mathcal{D}(v)}, \bigcup_{v \in \mathcal{V}_n} N_{\mathcal{D}(v)}$, and $\{n_G\}$ are all preserved under $\alpha$. Specifically, $N_{\mathcal{V}_p}, N_{\mathcal{V}_n}$, and $\bigcup_{v \in \mathcal{V}_p} N_{\mathcal{D}(v)}$ are preserved since, while being of the same color, each $n \in N_{\mathcal{V}_n}$ has zero neighbors of color 0, each $n \in N_{\mathcal{V}_p}$ has at least one neighbor of color 0 and an in-degree of magnitude zero, while $n \in \bigcup_{v \in \mathcal{V}_p} N_{\mathcal{D}(v)}$ has at

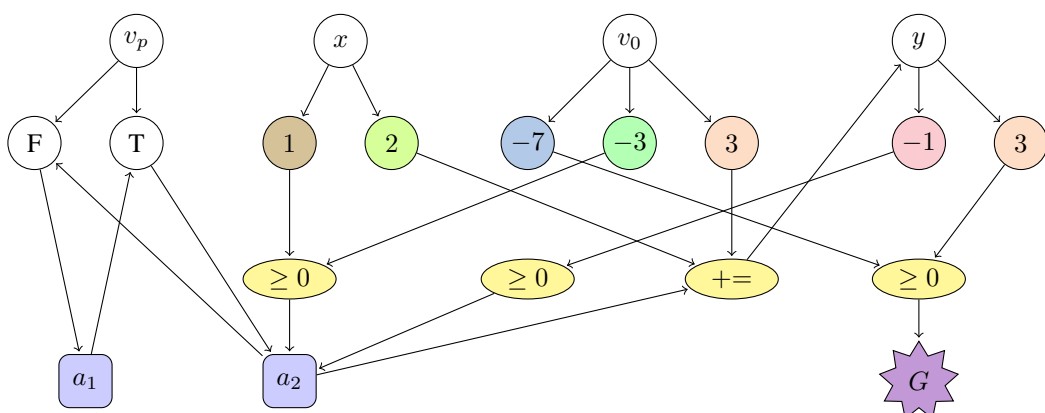

$NPDG_\Pi$

Figure 2: Toy example of a $NPDG_\Pi$ graph of a task $\Pi$, with the variable $\mathcal{V}_p = \{v_p\}$ and $\mathcal{V}_n = \{x, y\}$, actions $\mathcal{A} = \{a_1, a_2\}$, and the goal condition $G = \{3y \geq 7\}$. The actions $a_1$ and $a_2$ have the following form $\mathsf{pre}(a_1) = \{\langle v_p = F\rangle\}$, $\mathsf{pre}(a_2) = \{\langle v_p, T\rangle, x \geq 3, y \leq 0\}$, $\mathsf{eff}(a_1) = \{\langle v_p, T\rangle\}$, $\mathsf{eff}(a_2) = \{\langle v_p, F\rangle, y \mathrel{+}= 2x + 3\}$, and $\mathsf{cost}(a_1) = \mathsf{cost}(a_2) = 1$. The initial state is not given, since it is not involved in the construction of $NPDG$.

least one neighbor of color 0 and an in-degree that is at least one. All other sets differ due to different colors. Note also that since $\alpha$ is edge preserving we have that if $\alpha(n_v) = n_u$ then $\alpha(N_{\mathcal{D}(v)}) = N_{\mathcal{D}(u)}$. Vertices in $N_+$ must differ from the ones in $N_{\Psi_n}$ under $\alpha$, since each $n \in N_+$ has at least one outgoing neighbor of color 0, and the vertices in $N_{\Psi_n}$ has none of these. Hence, we have that $f(\alpha)$ obeys the properties 1-3 of the definition of NSS.

Property 5 is preserved also due to edge preservance and the fact that $\alpha(n_G) = n_G$. Note that if $\langle v, d\rangle \in G_p$ then, by construction of $NPDG_\Pi$, we have that $(n_d^v, n_G) \in E$, thus $(\alpha(n_d^v), \alpha(n_G) = n_G) \in E$, which means that $f(\alpha)(\langle v, d\rangle) \in G_p$. Now, let $\psi : \xi \geq 0 \in G_n$. Then, $(n_{\bar{\xi}}^{\geq}, n_G) \in E$, and once again $(\alpha(n_{\bar{\xi}}^{\geq}), n_G) \in E$. Note that for each $v \in vars(\xi)$ there is $w_v^\xi \in nums(\xi)$, and by construction of $NPDG_\Pi$ there are edges $(n_v, n_{w_v^\xi}^v), (n_{w_v^\xi}^v, n_{\bar{\xi}}^{\geq}) \in E$. Since $\alpha$ is edge preserving, we have the path $(\alpha(n_v), \alpha(n_{w_v^\xi}^v)), (\alpha(n_{w_v^\xi}^v), \alpha(n_{\bar{\xi}}^{\geq})), (\alpha(n_{\bar{\xi}}^{\geq}), n_G) \in E$, thus the summand $w_v^\xi f(\alpha)(v)$ is a part of the expression $f(\alpha)(\xi) \geq 0$, since $\alpha(n_{w_v^\xi}^v)$ and $n_{w_v^\xi}^v$ must be of the same color, $v$ and $f(\alpha)(v)$ have the same multiple in $\xi$ and $f(\alpha)(\xi)$, respectively. And since $\alpha$ is in-degree preserving, the number of summands in $\xi$ is equal to the number of summands in $f(\alpha)(\xi)$. Hence, $f(\alpha)(\xi)$ is well-defined in terms of NSS, and $f(\alpha)(\xi) \geq 0 \in G$. Therefore, $f(\alpha)(G) \subseteq G$. To show the converse we need to recall that $\alpha^{-1} \in Aut(NPDG_\Pi)$ and $f$ is a homomorphism.

To prove property 4, we need to show that $f(\alpha)(h(a)) = h(f(\alpha)(a))$ for each $h \in \{\mathsf{pre}_n, \mathsf{pre}_p, \mathsf{eff}_n, \mathsf{eff}_p, \mathsf{cost}\}$. This property on preconditions is obtained exactly as Property 5, where the node $n_G$ is replaced with the nodes $n_a$ and $n_{f(\alpha)(a)}$. The proof for effects is similar, but requires us to look at the nodes $n_\xi^+$, where the only outgoing edge is pointed towards a node that represents a numeric variable.

We omit the full proof here, since it repeats almost verbatim the one given in the previous paragraph.

For the **injectivity of** $f$, consider $\alpha, \beta \in Aut(NPDG_\Pi)$:

$$f(\alpha) = f(\beta) \implies l^{-1} \circ \alpha \circ l = l^{-1} \circ \beta \circ l \implies$$

$$\alpha(n) = \beta(n) \text{ for all } n \in N_\mathcal{V} \cup N_\mathcal{A} \cup \bigcup_{v \in \mathcal{V}_p} N_{\mathcal{D}(v)},$$

where the last implication follows from the fact that $l$ is bijective. To show that $f$ is injective, we need to show that $\alpha(n) = \beta(n)$ for all $n \in N$. Thus, assume in contradiction that there is $n \in N$ such that $\alpha(n) \neq \beta(n)$. These $n$ is either $n_G$ or lies in $N_\Xi \cup \bigcup_{v \in \mathcal{V}_n} N_{\mathcal{D}(v)}$. Since $n_G$ has its own color, it is a fixed under permutations in $Aut(NPDG_\Pi)$. Since $f$ is well-defined we have that $f(\alpha), f(\beta) \in Aut(\Pi)$, thus for $n_w^v \in \bigcup_{v \in \mathcal{V}_n} N_{\mathcal{D}(v)}$ we have

$$\alpha(n_w^v) = n_w^{f(\alpha)(v)} = n_w^{f(\beta)(v)} = \beta(n_w^v), \text{ and}$$

$$\alpha(n_\xi^\dagger) = n_{f(\alpha)(\xi)}^\dagger = n_{f(\beta)(\xi)}^\dagger = \beta(n_\xi^\dagger) \text{ for } n_\xi^\dagger \in N_\Xi.$$

Lastly, to show that $f$ **is surjective**, we need to prove that $f^{-1}(\sigma)$ is in $Aut(NPDG_\Pi)$, when $\sigma$ is an NSS. We extend this inverse as follows

$$f^{-1}(\sigma)(n) = \begin{cases} n_{\sigma(x)} & \text{if } n = n_x \in N_{\mathcal{V}_p} \cup N_{\mathcal{V}_n} \cup N_\mathcal{A} \\ n_{\sigma(\langle v,d\rangle)} & \text{if } n = n_d^v = n_{\langle v,d\rangle} \in \bigcup_{v \in \mathcal{V}_p}, \\ n_w^{\sigma(v)} & \text{if } n = n_w^v \in \bigcup_{v \in \mathcal{V}_n}, \\ n_{\sigma(\xi)}^\dagger & \text{if } n = n_\xi^\dagger \in N_\Xi \wedge \dagger \in \{+, \geq\}, \\ n_G & \text{if } n = n_G. \end{cases}$$

The extended function $f^{-1}(\sigma)$ is well-defined, i.e., all vertices that involve $\sigma$ in their indices do exist in $NPDG_\Pi$. To show this, we note that by definition $\sigma$ is a permutation on variables, actions, and $\sigma(\langle v, d\rangle) \in \mathcal{D}(\sigma(v))$. Moreover, for each $n_w^v$ there is $\xi \in \Xi$, such that $w \in nums(\xi)$. We saw that for an NSS it holds that $\sigma(\Xi) = \Xi$. Hence, $\sigma(\xi) \in \Xi$,

thus, $w\sigma(v)$ is a summand in $\sigma(\xi)$. Lastly, to show that $n^{\dagger}_{\sigma(\xi)} \in NPDG_{\Pi}$, note that $\sigma(\Xi) = \Xi$ and NSS preserves linear conditions and effects.

Next, note that, by construction, $f^{-1}$ is a homomorphism, since for $\sigma, \sigma' \in Aut(\Pi)$ it holds $f^{-1}(\sigma \circ \sigma') = f^{-1}(\sigma) \circ f^{-1}(\sigma')$. Moreover, for the identity element $\mathrm{id}_{\Pi} \in Aut(\Pi)$ it holds that $e := f^{-1}(\mathrm{id}_{\Pi})$ is an identity element in $Aut(NPDG_{\Pi})$. Let $\alpha := f^{-1}(\sigma)$, it is invertible, hence, a permutation since:

$$\alpha \circ f^{-1}(\sigma^{-1}) = f^{-1}(\sigma) \circ f^{-1}(\sigma^{-1}) = f^{-1}(\sigma \circ \sigma^{-1}) =$$
$$f^{-1}(\mathrm{id}_{\Pi}) = e \implies \alpha^{-1} = f^{-1}(\sigma^{-1}).$$

Next, we need to show that $\alpha$ is edge- and color-preserving. We begin with color preservation.

- $\alpha(N_{\mathcal{V}_p}) = N_{\mathcal{V}_p}$, $\alpha(N_{\mathcal{V}_n}) = N_{\mathcal{V}_n}$, and $\alpha(N_{\mathcal{A}}) = N_{\mathcal{A}}$ since $\sigma(\mathcal{V}_p) = \mathcal{V}_p$, $\sigma(\mathcal{V}_n) = \mathcal{V}_n$, and $\sigma(\mathcal{A}) = \mathcal{A}$, respectively. All vertices in $N_{\mathcal{V}}$ are of the same color. On action vertices, $n_a \in N_{\mathcal{A}}$, the colors are preserved due to
$$\mathsf{col}(\alpha(n_a)) = \mathsf{col}(n_{\sigma(a)}) = \mathsf{ord}(\mathsf{cost}(\sigma(a))) = $$
$$\mathsf{ord}(\mathsf{cost}(a)) = \mathsf{col}(n_a).$$
- For a variable $v \in \mathcal{V}_p$ we have $\sigma(\mathcal{D}(v)) = \mathcal{D}(\sigma(v))$. Thus, $\alpha(N_{\mathcal{D}(v)}) = N_{\mathcal{D}(\sigma(v))}$. Thus, $\alpha$ fixes the color on the propositional domains vertices.
- Let $v \in \mathcal{V}_n$, and $n_w^v \in N_{\mathcal{D}(v)}$, then
$$\mathsf{col}(\alpha(n_w^v)) = \mathsf{col}(n_w^{\sigma(v)}) = \mathsf{ord}(w) = \mathsf{col}(n_w^v).$$
- Note that for $n_\xi^{\dagger} \in N_{\Xi}$ it holds that $\alpha(n_\xi^{\dagger}) = n_{\sigma(\xi)}^{\dagger} \in N_{\Xi}$, and all vertices in $N_{\Xi}$ are of the same color.
- Trivially, $\mathsf{col}(\alpha(n_G)) = \mathsf{col}(n_G)$.

Thus, we have that $\alpha$ is color preserving vertex permutation. The last property for $\alpha$ being a color-preserving automorphism, is edge-preservance, $(n, n') \in E \iff (\alpha(n), \alpha(n')) \in E$.

The "$\Rightarrow$" part of this property follows from a meticulous application of NSS properties 1-5 to
$$E = E_{\mathcal{V}_p} \cup E_{\mathcal{V}_n} \cup E_{\Xi} \cup E_{\mathcal{A}} \cup E_G.$$
We omit the full proof here since it is straightforward and repetitive. Overall, the edges of $NPDG_{\Pi}$ correspond to element membership in sets, e.g., the edge $(n_d^v, n_G)$ corresponds to $\langle v, d \rangle \in G$, $(n_w^v, n_\xi^+)$ to $w_v \in nums(\xi)$ and $u \mathrel{+}= \xi \in \mathsf{eff}_n(a)$ for some $a \in \mathcal{A}$ and $v \in \mathcal{V}_n$. This part of the proof is similar to the proofs of Thm. 5 in Sievers et al. (2019) and Thm. 4.4 in Shleyfman (2020).

Briefly, the edges in $E_{\mathcal{V}_p}$ are preserved due to properties 1 and 2, $E_{\Xi}$ and $E_{\mathcal{A}}$ are preserved due to properties 3-4, and $E_G$ is preserved due to property 5. $E_{\Xi}$ is preserved since $\sigma(\Xi) = \Xi$. The "$\Leftarrow$" part holds since $\alpha$ is invertible. $\square$

To summarise, in this section we proved that the groups $Aut(NPDG_{\Pi})$ and $Aut(\Pi)$ are isomorphic, i.e., structurally identical, and that there is an injection that maps these groups into a subgroup of $Aut(\mathcal{T}_{\Pi})$, we name this group $\Gamma_{\Pi}$. The generators of $\Gamma_{\Pi}$ are the ones used by DKS and OSS for symmetry breaking. Note that if we have the generators of $Aut(NPDG_{\Pi})$, we can compute the generators of $\Gamma_{\Pi}$ in linear time. In the next section we discuss the computational complexity of computing the generators of $Aut(NPDG_{\Pi})$.

## Computational Complexity

The *graph isomorphism* (GI) decision problem that gets as input two finite graphs and determines if these graphs are isomorphic. This problem is neither known to be NP-complete nor tractable. Since many related problems appeared to be polynomial-time equivalent to the GI problem (Mathon 1979), it gave its name to a complexity class. As usual for complexity classes within the polynomial-time hierarchy, a problem $X$ is called GI-hard if there is a polynomial-time reduction from GI to $X$. A problem $X$ lies in GI if it can be reduced to the GI problem. A problem $X$ is GI-complete if it both lies in GI and GI-hard, i.e., polynomial-time equivalent to the GI problem.

Shleyfman (2019) showed that computing the generators for the groups of Structural Symmetries and the automorphisms of *PDG* is GI-hard. Since Numeric Planning contains FDR in the trivial case when $\mathcal{V}_n = \emptyset$, and in this case *NPDG* and *PDG* are equivalent, computing the *automorphism group of NPDG and NSS is also GI-hard*.

Shleyfman and Jonsson (2021), in turn, proved that computing the generators for the automorphism group of a colored is GI-complete. Since *NPDG* is a colored graph, the problem of computing generators of its automorphism group lies in GI. Thus, we have that *the computation of generators of NPDG and NSS is GI-complete*.

Note that while the computation on solutions to the GI problem is suspected to have a quasi-polynomial complexity (Babai 2015, 2016), there are some off-the-shelf symmetry detection packages such as *bliss* (Junttila and Kaski 2007) or *Saucy* (Darga, Sakallah, and Markov 2008), that while being a worst-case exponential in time, perform well in practice. For the experimental evaluation of the method proposed in this paper, we chose *bliss*.

## Experimental Evaluation

We implement the symmetry detection method in Numeric Fast Downward (Aldinger and Nebel 2017). All experiments are run on an Intel Xeon Gold 6148 processor with a 30-minutes time limit and 4 GB memory limit using GNU parallel (Tange 2011). We evaluate A*, DKS and OSS using the blind heuristic, operator-counting heuristic with LM-cut and state equations constraints, $h_{\mathrm{LP}}^{\mathrm{LM\text{-}cut,\,SEQ}}$ (Kuroiwa et al. 2021), for SCT domains, which is a subset of LT, and LM-cut heuristic adapted for LT planning, $h_2^{\mathrm{LM\text{-}cut}}$ (Kuroiwa, Shleyfman, and Beck 2022), for LT domains. We only show instances where symmetries are detected, since the overhead of detecting no symmetries is usually less than 5 sec.

Symmetries in classical planning vary from domain to domain, and where the GRIPPER has a symmetry group that is exponential in the number of balls in the rooms, the BLOCKSWORLD domain has no symmetries at all due to a specific order imposed on the blocks in the task. The efficiency of symmetry-breaking techniques can be attributed to the vast variety of domains that were introduced to classical planning, within the setting of International Planning Competitions that were held during the last three decades.

As a sanity check, we performed some preliminary experiments on the GRIPPER domain, where the balls have ho-

| | A* | | | DKS | | | OSS | | |
|---|---|---|---|---|---|---|---|---|---|
| | coverage | time | expansions | coverage | time | expansions | coverage | time | expansions |
| SCT | | | | Blind | | | | | |
| DEPOTS-SYM (20) | 4 | 18.6 | 1085229 | **5** | 11.2 | **305317** | **5** | **6.2** | 306229 |
| GARDENING (63) | 63 | 3.8 | 181199 | 63 | 4.2 | **103376** | 63 | **2.2** | 103405 |
| GARDENING-SAT (51) | 10 | 60.5 | 2082149 | **11** | 71.6 | **1261201** | **11** | **37.3** | 1265773 |
| DELIVERY (20) | 2 | 21.2 | 1174198 | 4 | 0.6 | **13717** | **6** | **0.3** | 13810 |
| ROVER (19) | 4 | 5.7 | 164822 | 4 | 4.2 | **70823** | 4 | **2.4** | 70825 |
| SAILING (20) | 0 | - | - | 0 | - | - | **1** | - | - |
| TOTAL (230) | 83 | - | - | 87 | - | - | **90** | - | - |
| Linear | | | | Blind | | | | | |
| ROVER-METRIC (10) | 4 | 4.4 | 154592 | 4 | 6.1 | 110913 | 4 | **3.2** | **110786** |
| TPP-METRIC (40) | 5 | **1.9** | 40406 | 5 | 4.1 | **39994** | 5 | 2.5 | 55768 |
| ZENOTRAVEL-LINEAR (9) | **3** | 1.1 | 25372 | 2 | 2.0 | **25190** | **3** | **1.0** | 25407 |
| BARMAN (15) | 2 | 0.7 | 32854 | 3 | 0.2 | **4388** | **4** | **0.1** | 4417 |
| BARMAN-UNIT (15) | 2 | 4.0 | 176334 | **3** | 0.6 | **12618** | **3** | **0.3** | 12708 |
| TOTAL (105) | 16 | - | - | 17 | - | - | **19** | - | - |
| SCT | | | | $h_{\mathrm{LP}}^{\mathrm{LM\text{-}cut,SEQ}}$ | | | | | |
| DEPOTS (6) | 1 | 1096.9 | 109254 | 1 | 831.7 | **87917** | 1 | **799.3** | 88237 |
| DEPOTS-SYM (20) | 6 | 257.4 | 80762 | 6 | **108.5** | **45422** | 7 | 148.6 | 61358 |
| GARDENING (63) | 63 | 3.7 | 17479 | 63 | 2.5 | 10546 | 63 | **2.3** | **10522** |
| GARDENING-SAT (51) | 12 | 66.6 | 298918 | 12 | 46.8 | **181075** | 12 | **40.9** | 181086 |
| DELIVERY (20) | 2 | 0.1 | 16 | **6** | **0.0** | **15** | **6** | **0.0** | 20 |
| ROVER (19) | 4 | 63.7 | 154971 | 4 | 30.4 | **66308** | 4 | **26.2** | 66909 |
| SAILING (20) | 20 | **0.4** | **109** | 20 | **0.4** | 109 | 20 | **0.4** | 116 |
| SAILING-SAT (30) | **7** | **0.5** | **178** | 6 | 0.6 | 178 | 6 | 0.6 | **178** |
| TOTAL (230) | 115 | - | - | 118 | - | - | **119** | - | - |
| Linear | | | | $h_2^{\mathrm{LM\text{-}cut}}$ | | | | | |
| FO-SAILING (16) | **1** | 631.6 | 188971 | 0 | - | - | **1** | **560.7** | **180080** |
| ROVER-METRIC (10) | 6 | 38.6 | 47845 | 6 | 47.6 | 58884 | 6 | **31.3** | **41070** |
| TPP-METRIC (40) | 5 | 8.0 | 11275 | 5 | **7.7** | **11209** | 5 | 8.9 | 15190 |
| ZENOTRAVEL-LINEAR (9) | 8 | 194.0 | 11261 | 8 | 132.4 | **8087** | 8 | **129.8** | 8628 |
| BARMAN (15) | 2 | 2.0 | 30287 | 3 | 0.3 | **4182** | **4** | **0.2** | 4208 |
| BARMAN-UNIT (15) | 2 | 161.5 | 9764 | 2 | **42.3** | **2592** | 2 | 84.2 | 2646 |
| TOTAL (105) | 24 | - | - | 24 | - | - | **26** | - | - |

Table 1: For the search time and the number of expansions, we took the average over instances solved by all of A*, DKS and OSS except for $h_2^{\mathrm{LM\text{-}cut}}$ in FO-SAILING, where A* and OSS solve one instance and DKS solves no instance.

mogeneous weights, and the gripper-robot has a maximum load. In this setting, the numeric approaches were consistent with the classical ones. From the 20 GRIPPER problems we have, $A^*, DKS, OSS$ equipped with blind heuristics solved $7, 20, 20$, respectively. The same algorithms equipped with numeric LM-cut solved $6, 20, 20$, in the same order. This results are in line with these reported by Domshlak et al. for classical planning. We do not include this results in Table 1, since we find these domains too simplistic.

Instead of GRIPPER we introduce a new domain DELIVERY. In this domain multiple robots equipped with multiple arms and a tray deliver objects in a building, represented by a digraph. In addition, we introduced a DEPOTS-SYM domain, a variation of the previously existing DEPOTS SCT domains, where the weights of the packages and the capacities of the trucks are "standardized" by replacing what seems to be arbitrarily chosen numbers by some standard upper bounds. For example, the packages were all given weights in the range $\{4, 6, 8, 10, 12\}$, while the truck got capacities in the range $\{10, 20, 40\}$. This allowed us to obtain a more symmetric domain while still having a meaningful numeric behavior. One may even argue that DEPOTS-SYM is closer to reality than DEPOTS due to standardization.

We also introduced a linear numeric version of the BARMAN domain. This domain comes with unit-cost actions, and actions that have costs only if a drink was poured from the dispenser. Interestingly enough, since the BARMAN domain has a lot of delete-effects, on the unit-cost version of the domain the blind version of OSS out-performs all other configurations, including those that use LM-cut.

One of the curious behaviors that can be spotted in Table 1, is that $OSS$ performs better than $DKS$. This was not the case on the classical domains, where the algorithms performed on par. We attribute this difference to the fact that $DKS$ stores both original and canonical states, while $OSS$ operates only on canonical states, and thus requires less memory. The fact that numeric states require significantly more memory may play a key role in this behavior.

## Conclusion

In this paper, we extended the notions of structural symmetries and problem description graphs from classical to numeric linear planning. This allowed us to use symmetry-breaking forward search techniques in the numeric setting.

Our experiments demonstrate that these techniques solve more problems than the current state-of-the-art. In the future, one may consider looking for a larger symmetry group.

## Acknowledgements

This work is partially supported by the Natural Sciences and Research Council of Canada. Alexander Shleyfman is also partially supported by the Israel Academy of Sciences and Humanities program for Israeli postdoctoral researchers.

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
