# OpenReview forum: "Symmetry Detection and Breaking in Cost-Optimal Numeric Planning"
_icaps-conference.org/ICAPS/2022/Workshop/HSDIP — HSDIP 2022_

### Official Review · Reviewer_D6SX · 2022-04-15
**Interesting work, worth being presented**

**Confidence:** 5
**Overall Score:** Accept

**Review:**

The paper extends the notion of structural symmetries to numeric planning, introducing a numeric problem description graph (NPDG) and showing that the automorphism of NPDG is a subgroup of the automorphisms of the state transition graph. This allows to use the new structural symmetries in DKS or OSS, exactly the same way as the old ones were used.
The experimental evaluation shows some advantage on existing and modified domains in terms of expanded nodes, which translates into a moderate coverage improvement.

The extension seems straightforward, but some details are necessary for correctness. I wish the authors would have given these details more space and discussion. For instance, The authors stabilize the weights, allocating a distinct color to each. Why was it done? Why can't the weights be permuted?
It is not clear why there is no connection between a linear expression in the effect and the variable on which that change is performed. Seems like the nodes in 6 need to have the connection to v, which should be reflected in the edges as well. In other words, it seems like you need to have a node per effect in addition to a node per action. If that's not the case, it would be great to explain why.

The experimental evaluation is quite limited, with some assertions made without experimental evidence. It would be great to measure just how much the better performance of OSS is due to better memory consumption and how much is it due to smaller number of expansions (in other words, better pruning). It would be interesting to see a plot of expansions comparison between A* and OSS/DKS. BTW, are these expansions before the last layer?

Still, the paper seems to be an interesting addition to the workshop and might spark a fruitful discussion and therefore I would like to see it presented.

Some questions to the authors:
Q1: How is the canonical representative computed? Are you using the same algorithm as for classical planning? If so, what's the order between states in numeric planning? In classical planning, it is a lexicographical order over the variable values, variables in the causal graph order. In numeric planning, what's the order between variables you are using?

Q2: Are the tested heuristics invariant under the new structural symmetries? If not, do you think that could be a source of the better performance of OSS over DKS?


Minor presentational comments and typos:
1. "The search exploits these classes by replacing all states in this class with some representative state. Domshlak et al. have shown that given a “path” where each consequent state was replaced by a representative state, one may efficiently reconstruct a corresponding path in the original state space. Hence, the expanded search tree must contain at most one representative of each class at all times."
These sentences talk about OSS, but not DKS, so it should be explicitly mentioned.
There are a few issues with this excerpt. Given a symmetry group, equivalence classes may be defined in multiple ways. The last sentence is correct (only?) when you define these equivalence classes through the representative state, i.e., two states are equivalent iff their representative state is the same. So, please consider rephrase this entire excerpt.
2. "In the later case" => "In the latter case.
3. "w_0 \in Q" you forgot the superscript.
4. Please change the notation for the partial state s^p to something that does not involve p. It is confusing, since you also use p to mark a propositional part of the planning problem.
5. Para starting with "Conditions can be either propositional or numeric": the second "propositional should have been numeric, if I am not mistaken. Also, "In this case, we say that ψ is satisfied by s if ψ[s] ≥ 0 holds." should probably be "In this case, we say that ψ is satisfied by s if ξ[s] ≥ 0 holds."
6. Please review the text in the definition of numeric effects.

---

### Official Review · Reviewer_VaVv · 2022-04-24
**Extension of Symmetry Pruning to Numeric Planning**

**Confidence:** 3
**Overall Score:** Accept

**Review:**

# Summary

Numeric Planning is an extension from classical planning with additionally numeric variables, conditions which use those variables, and effects which modify those variables. The authors extend symmetries from classical planning to numeric planning, and propose an extension for the PDG to numeric planning such that it detects symmetries in numeric planning. Finally, they compare the performance of A* with and without symmetry pruning on numeric planning tasks.

# Feedback

The background is well introduced. I learned how numerical planning tasks are defined and got a fresh reminder on structural symmetries and the PDG. The definition of numeric structural symmetries makes intuitively sense. Your explanation for properties 1,3,5 confused me more than it helped me. The extended PDG is also reasonable for me. I admit that I was not able to follow the proofs completely. Figure 1 was helpful for me.

The results look very promising, but this is a bit hidden by the presentation. I suggest to sort the columns first by the metrics (coverage, time, expansions) and then within each metric by A*, DKS, OSS. This allows to easily compare the coverage/time/expansions between the algorithms, as they are just side by side. I would repeat in the main text that the expansion and time averages are taken over the commonly solved instances, and I would like an average across domains (weighted such that each domain has the same impact). Furthermore, you did not discuss your results at all. I strongly suggest to add a section for this.

As minor comments
- References: For some ICAPS publications the page numbers are missing.
- General: You use at least twice erroneous inline citations. In the introduction, `Domshlak et al.` is missing the year and the preliminaries the name `Wehrle` is misspelled as `Wherle`. I suggest to use `\citet{bibkey}` for inlinecitations.
- Preliminaries: "We say that $\psi$ is satisfied by $s$, ..., if $\psi\subseteq s$." I might be clearer if you use "$\psi \subseteq s_p$".
- Numeric Problem Description Graph:
	- If you have the space, e explicitly with the node explanation `1.` and write it down once for each set (p, n).
	- I would have been useful to remind me again of the meaning of $\Psi$.
	- \noindent before the definition of the edges?

---

### Author Response · Authors · 2022-04-28
**Response to the Reviews**

We would like to thank the reviewers for their hard work and helpful comments.

R1.
Adding a Discussion section is a great idea, unfortunately, we are currently pushing the space limit. The only way to get more space is to move the proofs to the supplementary materials, which will result in a heavy reconstruction of the paper, and submitting a partially unreviewed camera-ready. Thus, we would postpone this reorganization for the submission to another venue. Thank you for the constructive suggestions.


R2.
Seems like the nodes in 6 need to have the connection to v, which should be reflected in the edges as well.

You are right, we missed the edges $(n^{+}_{\xi}, n_v)$ in the definition of the NPDG. Thank you for spotting this one. Note that these edges do appear in Figure 2, and is mentioned in the proof of Theorem 3 (same page, top of the second column).

Q0: It would be interesting to see a plot of expansions comparison between A* and OSS/DKS. BTW, are these expansions before the last layer?

A0: No. We report all expanded states, including the last layer.

Q1: How is the canonical representative computed? Are you using the same algorithm as for classical planning? If so, what's the order between states in numeric planning? In classical planning, it is a lexicographical order over the variable values, variables in the causal graph order. In numeric planning, what's the order between variables you are using?

A1: As in classical planning, we use the lexicographic order on the variables. We first compare the propositional state variables, and then the numeric variables. The order of the numeric variables is decided by NFD.

Q2: Are the tested heuristics invariant under the new structural symmetries? If not, do you think that could be a source of the better performance of OSS over DKS?

A2: Blind heuristic is symmetric, while LM-cut is not in the direct sense. One can probably prove that LM-cut is symmetric in expectation, as it is in the classical case. That said, we believe that the symmetry of the heuristic has little to do with the fact that OSS overperforms DKS coverage-wise. Note that DKS consistently expands fewer nodes than OSS. The main problem of DKS, however, is that its states are twice the size of the ones kept by OSS. In comparison to classical planning, numeric planning has larger states (storing real numbers is a bit more expensive) and fewer symmetries. This lack of overhead makes the OSS slightly more competitive than DKS. We believe that if we find a way to produce larger symmetry groups, or apply both algorithms to highly symmetric domains, DKS might outperform OSS.

---

> ### Comment · Reviewer_D6SX · 2022-04-29
> **Order of numeric variables?**
>
> Thank you for your clarifications.
>
> I would suggest to account for expansions before the last layer, as the order in the last layer is somewhat arbitrary.
>
> You haven't answered what is the order of numeric variables, how is that computed by the numeric planner. The variable order for canonical state computation might have huge effect on the orbit size. To give you an anecdotal evidence, reverting the order in the classical planning case in gripper domain captures very little of the problem symmetries, while using the causal graph ordering allows to easily solve any gripper task that can be grounded.

---

> > ### Author Response · Authors · 2022-04-29
> > **Order of variables by NFD**
> >
> > In the implementation, the order of propositional/numeric variables is the causal graph ordering computed by preprocessing.